# Changes in countermovement jump force-time characteristic in elite male basketball players: A season-long analyses

**Nicolas M. Philipp***, **Dimitrije Cabarkapa, Ramsey M. Nijem, Andrew C. Fry**

Jayhawk Athletic Performance Laboratory - Wu Tsai Human Performance Alliance, University of Kansas, Lawrence, Kansas, United States of America

\* nicophilipp@ku.edu

## Abstract

Basketball is a sport that is characterized by various physical performance parameters and motor abilities such as speed, strength, and endurance, which are all underpinned by an athlete's efficient use of the stretch-shortening cycle (SSC). A common assessment to measure SSC efficiency is the countermovement jump (CMJ). When performed on a force plate, a plethora of different force-time metrics may be gleaned from the jump task, reflecting neuromuscular performance characteristics. The aim of this study was to investigate how different CMJ force-time characteristics change across different parts of the athletic year, within a sample of elite collegiate male basketball players. Twelve basketball players performed CMJ's on near-weekly basis, combining for a total of 219 screenings. The span of testing was broken down into four periods: pre-season, non-conference competitive period, conference competitive period, and post-season competitive period. Results suggest that basketball players were able to experience improvements and maintenance of performance with regards to various force-time metrics, transitioning from the pre-season period into respective later phases of the in-season period. A common theme was a significant improvement between the pre-season period and the non-conference period. Various force-time metrics were subject to change, while outcome metrics such as jump height remained unchanged, suggesting that practitioners are encouraged to more closely monitor how different force-time characteristics change over extended periods of time.

## Introduction

Basketball is arguably one of the most popular sports within the United States of America and is continuing to gain popularity around the globe [1]. The nature of basketball gameplay is characterized by athletes being exposed to many different physical performance tasks. According to Schelling and Torres-Ronda [2], success within basketball from both a technical and tactical standpoint requires athletes to display proficiency within a vast array of physical performance parameters and motor abilities such as speed, strength, and endurance. Given the nature of basketball gameplay, athletes frequently perform high intensity accelerations, decelerations, changes in direction, as well as vertical jumps [3–5].

**Data Availability Statement:** The minimal dataset is available on the open science framework (link provided) osf.io/bq7n9.

**Funding:** The authors received no specific funding for this work.

**Competing interests:** The authors have declared that no competing interests exist.

Within the available basketball literature, some studies have aimed to quantify longitudinal changes across different physical performance qualities. For instance, Heishman et al. [6] highlighted that across the pre-season period, National Collegiate Athletic Association (NCAA) Division-I male basketball players experienced moderate decreases in CMJ force-time metrics, paralleled with increases in external training loads. However, changes in neuromuscular performance in form of CMJ force-time characteristics were not reported across later phases within the in-season period. Further, some authors have looked at hormonal changes over the course of a basketball season [7, 8]. While more of such studies, utilizing longitudinal study designs are surfacing within the sport science literature, scientists and strength and conditioning practitioners still reported a lack of longitudinal data on high-level athletes [9, 10]. For instance, Bishop [11] has proposed that the observation of athletes in real world settings, specifically over extended periods of time, may be beneficial to sport science practitioners aiming to better understand how to optimize their approach towards performance within a given sport. Thus, practitioners might find it valuable to gain further insights into how neuromuscular performance changes across the entirety of the basketball season.

Based on the commonly seen movements tasks in basketball, it is reasonable to suggest that success within these physical performance parameters is largely underpinned by an athlete's efficient use of the stretch-shortening cycle (SSC). The SSC is a neuromuscular phenomenon experienced during the vast majority of dynamic tasks such as running or jumping. By definition, the SSC describes a phenomenon consisting of an eccentric phase or stretch followed by an isometric transitional period (amortization phase), leading into an explosive concentric action [12]. One commonly utilized physical performance assessment to measure athlete's SSC efficiency and overall neuromuscular function is the vertical countermovement jump (CMJ) [13]. More specifically, the CMJ may be broken down into different phases such as unweighing, braking, deceleration, concentric, flight, and landing phase. A deeper analysis of different force-time characteristics within the CMJ subphases enables practitioners to paint a clearer picture of SSC efficiency and neuromuscular function, beyond mere jump height. Kinetic data related to these force-time characteristics are commonly extracted from force-platform technology often referred to as force plates. According to Schuster et al. [14] force platforms have become a central tool in screening, profiling, monitoring, and rehabilitating elite athletes. In sports such as basketball in which irregular and congested competitive schedules are a common theme, force platforms, specifically the implementation of CMJ's on force platforms, has been shown to be a commonly utilized tool for monitoring athlete neuromuscular performance [14]. Longitudinal monitoring of CMJ force-time characteristics may provide practitioners with actionable insights into their athlete's fatigue and "readiness" status [13]. Further, it may serve as a tool to evaluate the effectiveness of different training modalities (improvement vs. regression), in addition to performance changes across different parts of the athletic year, likely brought about by variations in training and competition volume and density.

With the previously highlighted points in mind, and primarily due to the lack of longitudinal research within high-level athletes, the aim of the present study was to monitor and analyze changes in CMJ force-time characteristics across multiple phases of the athletic year (i.e., pre-season, non-conference, conference, and post-season) within a sample of NCAA Division-I elite male basketball players. It was hypothesized that CMJ force-time characteristics may change across different phases of the athletic year, likely brought upon by variations in the volume and density of training and competition schedules.

## Materials and methods

### Experimental approach to the problem

A time-series research design was implemented in the present study. Data was collected on a near-weekly basis as part of the team's regular strength and conditioning sessions. For the sake of this study, the span of testing was broken down into four periods: pre-season, non-conference, conference, and post-season period (Table 1). In Table 1, test days refer to the total amount of days on which CMJ's were performed, while athlete screenings refer to the total amount of data points collected (mean of three jumps). Prior to collection of jump data, all athletes were exposed to a standardized dynamic warm-up protocol led by the team's certified strength and conditioning coach. The warm-up procedure consisted of a number of dynamic stretching exercises (e.g., high-knees, butt-kicks, lunges, A-skips), and 2–3 practice vertical CMJ performed at the beginning of each respective training session.

### Subjects

The sample size for this investigation consisted of 12 elite NCAA Division-I male basketball players (age = 20.3 ± 2.1 years, weight = 94.5 ± 12.3 kg, height = 196.6 ± 10.2 cm). All participants were free of musculoskeletal injuries and cleared for participation in the team's training activities by the respective sports medicine staff. All testing procedures performed in the present study were approved by the University of Kansas's Institutional Review Committee and all subjects signed an Informed Consent Form (STUDY00148265).

### Countermovement jump testing

Following a dynamic warm-up, who's execution stayed consistent over the course of the study, athletes performed a total of three CMJs. Each jump was separated by 30 seconds rest interval to minimize a possible influence of fatigue. Data was recorded prior to the teams sport practice session, using the ForceDecks Dual Force Platforms (Vald Performance, Brisbane, Australia). Force platforms were zeroed prior to each data collection. Athletes were instructed to step onto the force plate and stand still with their hands on the hips for 2–3 seconds. Then, they were asked to jump as fast and as high as possible, while keeping their hands on the hips. Strong verbal encouragement was provided to ensure that maximal effort was given during each jump.

For the sake of this study, the start of the unloading phase was defined when the athlete's total force was reduced by more than 20 Newtons from baseline system mass and ended at minimum force recorded during the eccentric phase of CMJ, as suggested by the manufacturer. The eccentric phase was defined as the phase containing negative velocity. The eccentric braking phase was defined as a sub-phase of the eccentric phase, starting at minimum force, until the end of the eccentric phase. The deceleration phase was defined as another sub-phase of the eccentric phase, from peak eccentric velocity until the end of the eccentric phase.

**Table 1. Descriptive statistics for different periods of season.**

| Competitive Period | Test Days | Athlete Screenings | Games Played |
|---|---|---|---|
| Pre-Season | 7 | 57 | 0 |
| Non-Conference | 6 | 65 | 15 |
| Conference | 6 | 68 | 18 |
| Post-Season | 3 | 29 | 9 |

**Table 2. List and definition of force-time metric examined in the present study.**

| Strategy Metrics (unit) | Definition |
|---|---|
| Braking Phase Duration (s) | Duration of the braking phase |
| CON Phase Duration (s) | Duration of the CON phase |
| Countermovement Depth (cm) | Lowest center of mass displacement, transition from ECC to CON phase |
| ECC Deceleration Phase Duration (s) | Duration of the ECC deceleration phase |
| ECC Peak Velocity (m•s$^{-1}$) | Maximal velocity obtained during the ECC phase |
| **Driver Metrics (unit)** | **Definition** |
| CON Impulse (N•s) | Area under the CON phase of the net force-time curve |
| CON Mean Force (N) | Average force of the CON phase |
| CON Peak Force (N) | Peak force of the CON phase |
| CON RFD (N•s$^{-1}$) | The average change in force over time during the CON phase |
| ECC Braking Impulse (N•s) | Area under the ECC braking phase of the net force-time curve |
| ECC Braking RFD (N•s$^{-1}$) | Average change in force over time during ECC braking time |
| ECC:CON Mean Force Ratio (%) | Ratio of mean forces in the ECC and CON phases |
| ECC Deceleration Impulse (N•s) | Area under the ECC deceleration phase of the net force-time curve |
| ECC Deceleration RFD (N•s$^{-1}$) | The average change in force over time during the ECC deceleration phase |
| ECC Mean Braking Force (N) | Average force generated during the ECC braking phase |
| ECC Mean Deceleration Force (N) | Average force generated during the ECC deceleration phase |
| ECC Peak Force (N) | Peak force of the ECC phase |
| Force at Zero Velocity (N) | Total force at the instance velocity is zero prior to take-off |
| Unloading Impulse (N•s) | Net impulse from start of movement to start of deceleration phase |
| **Outcome Metrics (unit)** | **Definition** |
| Jump Height (cm) | Maximal jump height via impulse—momentum calculation |
| RSI-modified (ratio) | Jump height divided by contraction time |

Note: RFD = rate of force development; ECC = eccentric; CON = concentric; RSI = reactive strength index

Performance metrics of interest were further classified as being either strategy, driver, or outcome metrics [15].

On individual testing days, the mean of the three jump trials was calculated for respective metrics of interest [16]. Force-time metrics used within these analyses are presented in Table 2. Force-time metric definitions were adapted from Merrigan et al. [17]. Concentric rate of force development was excluded from any further analyses, given its less reliable nature, identified during the first two weeks of testing, and highlighted within previous research [18, 19].

## Statistical analyses

All data were checked for normality using the Shapiro-Wilk statistic. Given the real-world nature, and size of this sample, statistical outliers were not removed prior to further analyses [20]. Within session coefficients of variation (CV) for all metrics of interest were calculated during the first two weeks of testing, to ensure reliability of variables. Linear mixed models were used to investigate mean differences in primary study outcomes (e.g., different CMJ force-time metrics) across the fixed factor of time (e.g., pre-season vs. non-conference), using the individual athlete as a random factor. All post-hoc comparisons were adjusted using the Bonferroni correction. Statistical inferences were made using an alpha level of $p \leq 0.05$. All data were analyzed using the R statistical computing environment and language (v. 4.0; R Core Team, 2020) via the Jamovi graphical user interface. Data were further visualized using the RStudio Software (Version 1.4.1106).

## Results

Average within session CVs for all metrics of interest, except for concentric rate of force development ranged from 2–12% and were therefore deemed acceptable. Concentric rate of force development demonstrated an average within session CV of 64% during the first two weeks of testing and was therefore eliminated from further analyses.

Within the group of strategy metrics, significant univariate effects for period-specific changes in braking phase duration were observed (F = 4.46, p = 0.005). Specifically, athletes performed a shorter braking phase duration during the non-conference period, compared to the pre-season period (p = 0.003). Similar results were observed looking at the deceleration phase duration (F = 4.92, p = 0.003), and concentric phase duration (F = 5.79, p = <0.001), with athletes performing a significantly shorter deceleration phase (p = 0.002), and concentric phase during the non-conference period, compared to the pre-season period (p < 0.001). Countermovement depth also revealed significant effects for period-specific changes (F = 3.92, p = 0.009), with athletes performing a significantly shallower countermovement during the non-conference period, compared to the pre-season period (p = 0.009). No significant differences between period were observed for eccentric peak velocity.

Within the group of driver metrics, significant univariate effects for period-specific changes in concentric mean force (F = 3.62, p = 0.014), eccentric braking rate of force development (RFD) (F = 3.17, p = 0.025), eccentric deceleration RFD (F = 2.97, p = 0.033), eccentric:concentric mean force ratios (F = 3.22, p = 0.024), and eccentric peak force (F = 3.45, p = 0.018) were observed. More specifically, and similar to findings from the strategy metric group, athletes generated significantly larger amounts of concentric mean force (p = 0.012), braking RFD (p = 0.018), deceleration RFD (p = 0.035), and eccentric mean deceleration force (p = 0.012) during the non-conference period, compared to the pre-season period. Additionally, athletes showed significantly lower eccentric:concentric mean force ratios during the non-conference period, compared to the pre-season period (p = 0.021). Moreover, looking at eccentric mean braking force, athletes generated significantly larger magnitudes of force during the post-season period (p = 0.037), as well as conference period (p = 0.033), compared to the pre-season period. Lastly, athletes generated significantly larger peak eccentric forces during the conference period, compared to the pre-season period (p = 0.045). No significant period-specific changes were observed for concentric impulse, force at zero velocity, as well as eccentric braking and deceleration impulse. Finally, within the group of outcome metrics, no significant univariate effects for period-specific changes in jump height or Reactive Strength Index (RSI)-modified were observed. Table 3 shows descriptive statistics for countermovement jump metrics across the four time periods analyzed in this study. Figs 1–5 visualize changes for respective metrics that reached statistical significance, across the span of the season.

## Discussion

The purpose of the present study was to monitor and analyze changes in CMJ force-time characteristics across multiple phases of the athletic year (i.e., pre-season, non-conference, conference, and post-season competitive periods) within a sample of NCAA Division-I elite male basketball players. It was hypothesized that CMJ force-time characteristics may change across different phases of the athletic year, likely brought upon by variations in the volume and density of training and competition schedules. In line with our hypothesis, between-period changes in CMJ force-time characteristics were observed for the group of athletes within our investigation. Values from both the groups of strategy metrics, and driver metrics were subject to change, with a common trend being an improvement in performance from the pre-season to the non-conference training period. From a statistical standpoint, outcome metrics

**Table 3. Differences in countermovement jump metrics across the four time periods.**

| Strategy Metrics (unit) | Pre-Season | Non-Conference | Conference | Post-Season |
|---|---|---|---|---|
| Braking Phase Duration (s) | 0.289 ± 0.042 | **0.271 ± 0.029\*** | 0.279 ± 0.034 | 0.273 ± 0.023 |
| CON Phase Duration (s) | 0.246 ± 0.031 | **0.234 ± 0.024\*** | 0.237 ± 0.030 | 0.238 ± 0.025 |
| Countermovement Depth (cm) | -31.9 ± 4.38 | **-30.2 ± 3.96\*** | -31.2 ± 4.43 | -30.8 ± 3.88 |
| ECC Deceleration Phase Duration (s) | 0.165 ± 0.029 | **0.154 ± 0.022\*** | 0.156 ± 0.022 | 0.154 ± 0.018 |
| ECC Peak Velocity (m•s⁻¹) | -1.35 ± 0.137 | -1.36 ± 0.122 | -1.38 ± 0.155 | -1.37 ± 0.138 |
| **Driver Metrics (unit)** | | | | |
| CON Impulse (N•s) | 268 ± 32.7 | 268 ± 34.6 | 265 ± 32.9 | 264 ± 33.0 |
| CON Mean Force (N) | 2027 ± 182 | **2085 ± 214\*** | 2045 ± 183 | 2033 ± 222 |
| CON Peak Force (N) | 2459 ± 221 | **2545 ± 260\*** | **2519 ± 209\*** | **2504 ± 259\*** |
| ECC Braking Impulse (N•s) | 67.8 ± 14.1 | 68.6 ± 16.1 | 67.1 ± 14.5 | 68.4 ± 15.6 |
| ECC Braking RFD (N•s⁻¹) | 7528 ± 1918 | **8330 ± 1669\*** | 8030 ± 1551 | 8043 ± 1424 |
| ECC:CON Mean Force Ratio (%) | 46.0 ± 4.13 | **45.0 ± 3.48\*** | 45.0 ± 4.55 | 45.3 ± 3.51 |
| ECC Deceleration Impulse (N•s) | 128 ± 24.4 | 131 ± 24.2 | 130 ± 24.5 | 129 ± 22.3 |
| ECC Deceleration RFD (N•s⁻¹) | 8790 ± 3087 | **9873 ± 3341\*** | 9675 ± 2987 | 9426 ± 2200 |
| ECC Mean Braking Force (N) | 1173 ± 161 | 1194 ± 174 | 1164 ± 150 | **1172 ± 150\*** |
| ECC Mean Deceleration Force (N) | 1729 ± 229 | **1804 ± 240\*** | **1767 ± 200\*** | **1760 ± 214\*** |
| ECC Peak Force (N) | 2293 ± 259 | 2380 ± 282 | **2350 ± 237\*** | 2334 ± 281 |
| Force at Zero Velocity (N) | 2289 ± 260 | 2372 ± 282 | 2344 ± 236 | 2327 ± 285 |
| Unloading Impulse (N•s) | -129 ± 24.4 | -131 ± 24.1 | -130 ± 24.5 | -129 ± 22.3 |
| **Outcome Metrics (unit)** | | | | |
| Jump Height (cm) | 41.5 ± 7.53 | 40.8 ± 6.40 | 41.7 ± 7.58 | 41.2 ± 7.45 |
| RSI-modified (ratio) | 0.62 ± 0.11 | 0.64 ± 0.10 | 0.65 ± 0.13 | 0.63 ± 0.11 |

*Note: "*" = significantly different from pre-season value

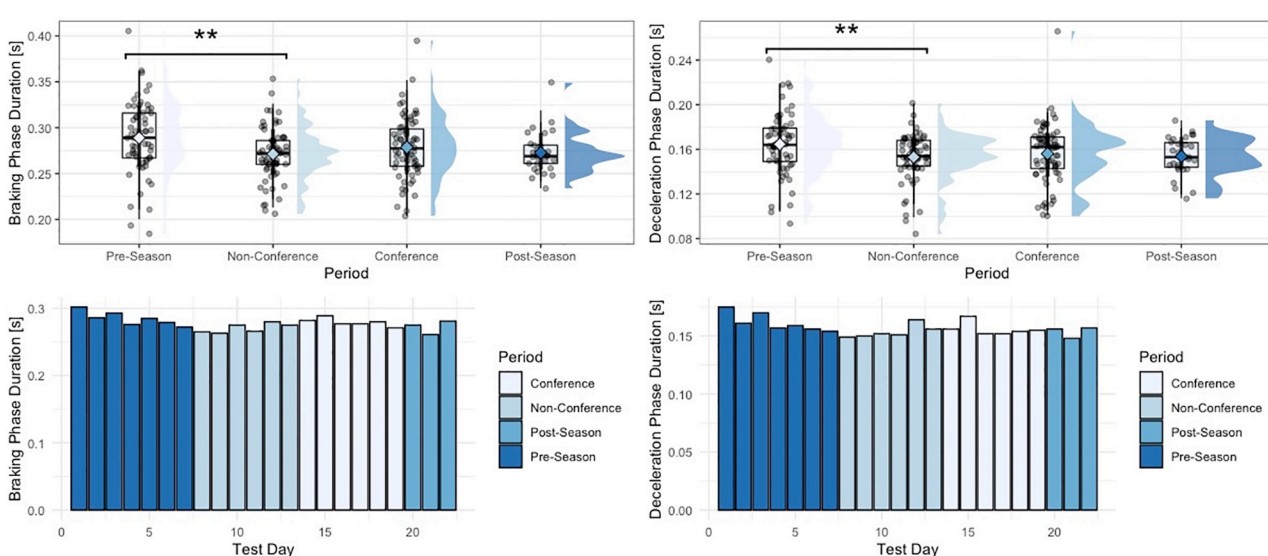

**Fig 1. Changes in braking and deceleration phase duration across the span of a season.** * "**" = p-value < 0.01, "*" = p-value < 0.05, phase-specific raincloud plots include boxplots with interquartile range and whiskers, as well as individual data density.

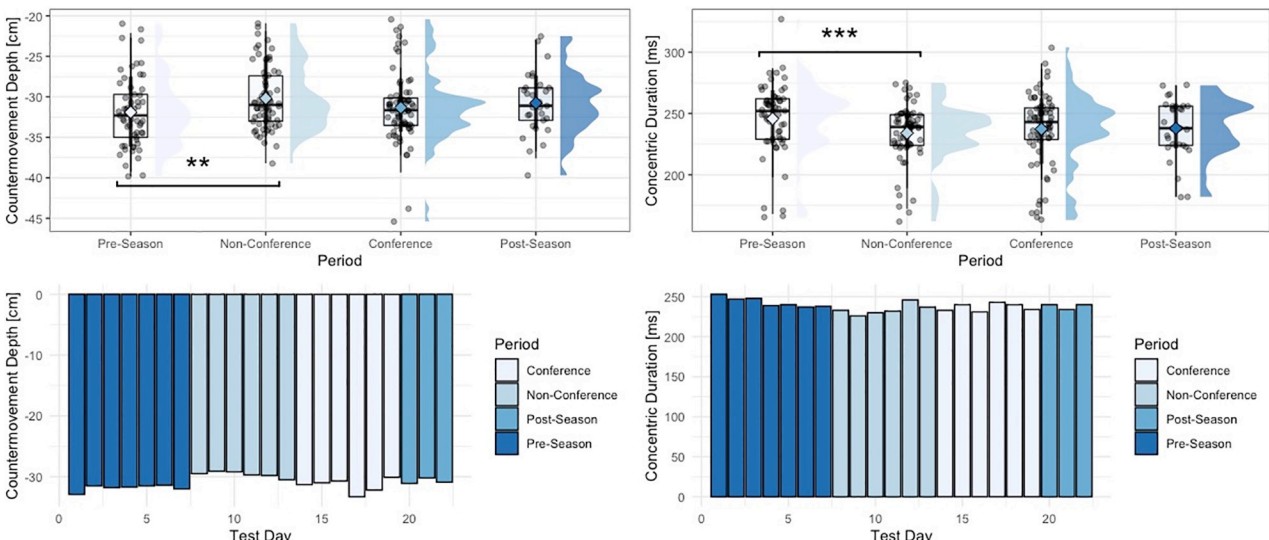

**Fig 2. Changes in countermovement depth and concentric duration across the span of a season.** "**" = p-value < 0.01, "****" = p-value < 0.001, phase-specific raincloud plots include boxplots with interquartile range and whiskers, as well as individual data density.

remained unchanged across the four periods. Therefore, a first potential actionable take-away, based on our findings, might be that solely measuring outcome measures such as jump height may be short-sighted when monitoring CMJ performance changes across an extended period of time within male basketball athletes. Interestingly, Luebbers et al., found reductions in vertical jump height in later phases of the season, compared to the pre-season, within a sample of female basketball players [8]. On the contrary, and within a similar research design, Matulaitis et al. [21] suggested that elite youth male basketball players experienced significant improvements in CMJ jump height and lane agility completion times between the preparatory period,

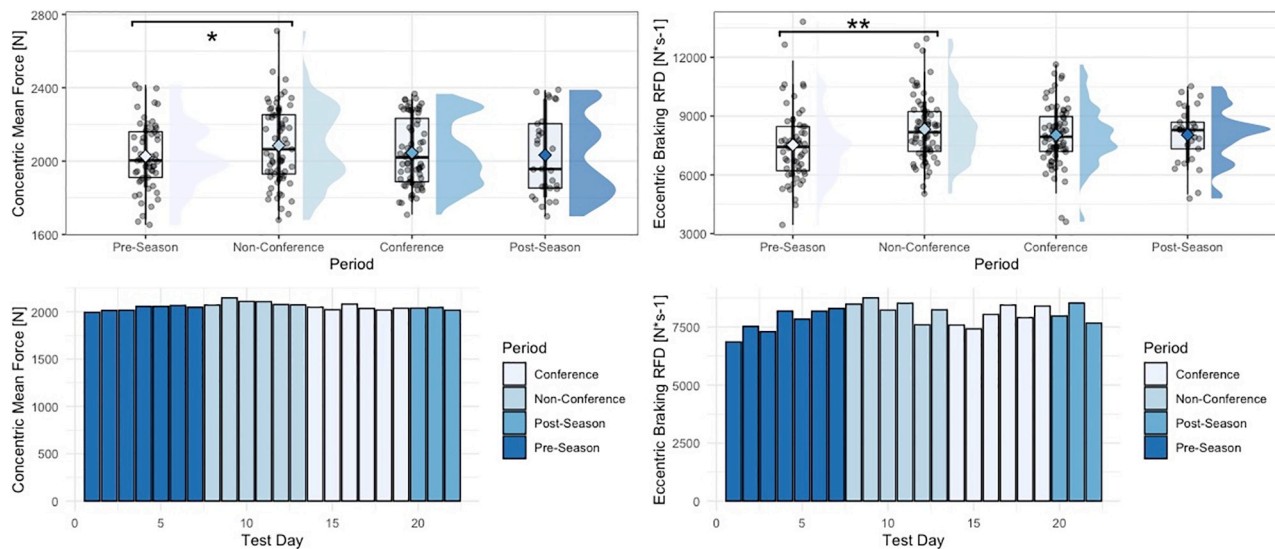

**Fig 3. Changes in concentric mean force and eccentric braking rate of force development across the span of a season.** "**" = p-value < 0.01, "*" = p-value < 0.05, phase-specific raincloud plots include boxplots with interquartile range and whiskers, as well as individual data density.

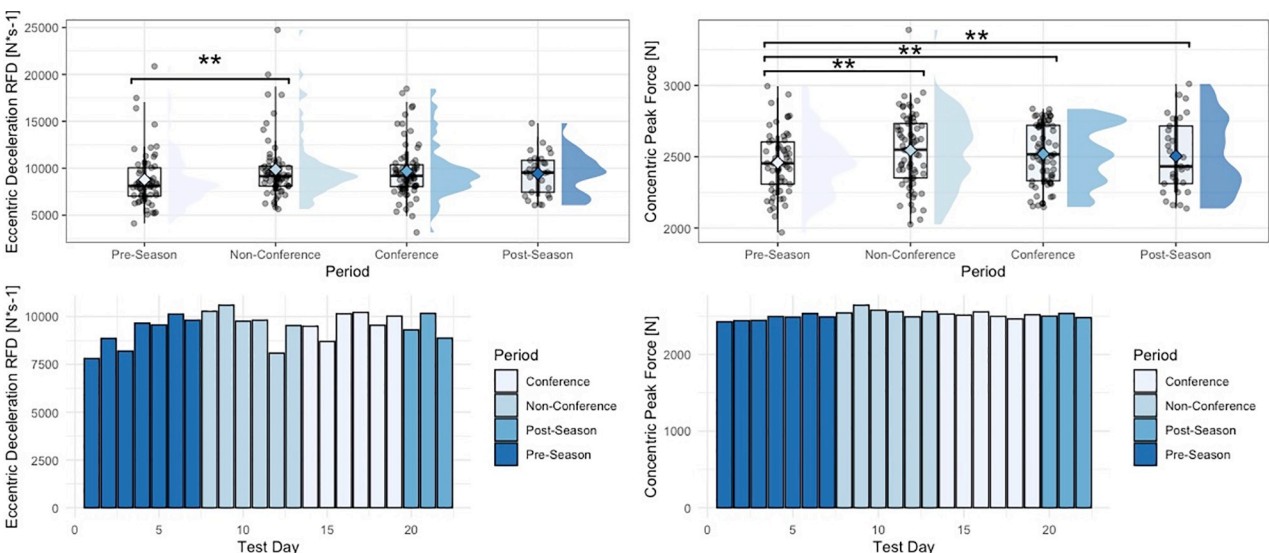

**Fig 4. Changes in eccentric deceleration rate of force development and concentric peak force across the span of a season.** "**" = p-value < 0.01, phase-specific raincloud plots include boxplots with interquartile range and whiskers, as well as individual data density.

and the second competitive period of the season. Similarly, Cruz et al. [22] suggested that over a nine-week competitive period, female basketball players experienced improvements in CMJ heights, despite vast variations in weekly training loads. However, these studies did not take into consideration underlying force-time characteristics that influenced the achievement of respective jump heights. Furthermore, within the present study, primary CMJ performance improvements were observed between the pre-season and the non-conference period, with a maintenance in performance experienced across the later periods of the season. Compared to the pre-season period, athletes performed a faster, shallower, and more forceful

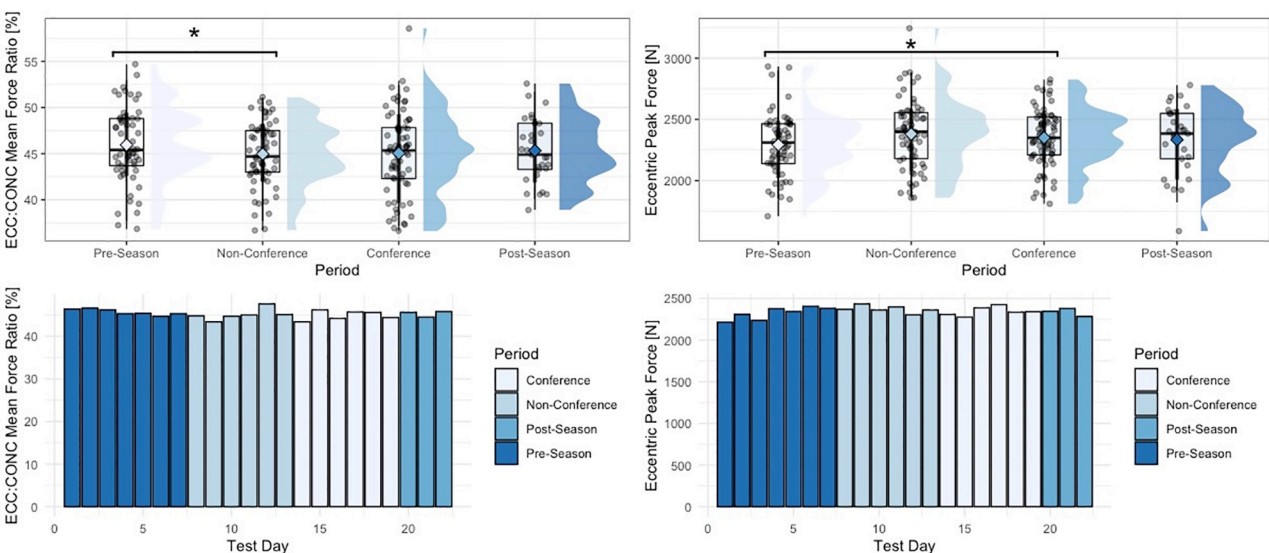

**Fig 5. Changes in eccentric:concentric mean force ratio and eccentric peak force across the span of a season.** "*" = p-value < 0.05, RFD = Rate of Force Development, phase-specific raincloud plots include boxplots with interquartile range and whiskers, as well as individual data density.

countermovement. Acutely, this might have been a supercompensation adaption induced by pre-season periodization strategies, in conjunction with an absence of games experienced during this period. Additionally, this could have also been attributed to the learning curve associated with performing CMJ's, by those athletes that have previously not performed this movement. However, during the pre-season training period, athletes are commonly exposed to intensified forms of training, which might support the finding of lower performance within respective force-time metrics. Both increases and decreases in CMJ performance across the pre-season period in basketball players have been highlighted within the previous literature [23, 24]. Further, similar findings were highlighted by Gonzalez et al. [25] who found that NCAA Division-I female basketball players were able to maintain or improve vertical jump power across an athletic season. More interestingly, findings suggested that those athletes who were starters experienced significantly greater improvements in vertical jump power, compared to the group of non-starters, despite greater decreases in subjective measures of energy, focus, and alertness. Contrary to popular belief, this may suggest that greater playing time may have acted as a stimulus to increase vertical jump power [3]. Within the realms of our study, this may help explain improvements in CMJ force-time metrics that were experienced between the pre-season, and non-conference period, and were maintained throughout the end of the season. According to Petway et al. [3], higher-level basketball players, such as the ones examined in the present study, seem to present with greater movement efficiency on the court. When compared with lower-level and youth players, high-level basketball players tend to cover less distance at lower average velocities and with lower average and maximal heart rates during competition [3]. These factors may also help explain why athletes within our sample were able to experience performance increases and maintenance across different phases of the athletic year, with regards to different CMJ force-time characteristics.

As previously mentioned, significant period-specific changes were seen within the group of strategy and driver metrics, while outcome metrics remained unchanged. Outcomes such as jump height or RSI-modified are influenced by several different force and time-dependent variables that underpin how well an athlete may jump. Therefore, if respective technology is available, practitioners should aim to analyze and monitor changes in neuromuscular performance more closely.

While novel, our study is not without limitations. Largely impacted by the uncontrollable nature of collegiate sports, particularly in-season, researchers were unable to control for factors such as weekly training or game loads, as well as nutritional intake or sleep schedules. Furthermore, information about specific sport practices, and strength and conditioning sessions across the span of the study were not taken into consideration within this study. Future investigations of this nature should aim to control for factors such as the ones mentioned above.

## Conclusion

In summary, the findings of the present study suggest that collegiate male basketball players were able to experience improvements and maintenance of performance with regards to various CMJ force-time metrics, transitioning from the pre-season period, into respective later phases of the in-season period. A common theme was a significant improvement between the pre-season period and the non-conference period. Primary study implications suggest that merely monitoring outcome metrics such as jump height may fail to paint a complete picture of how athlete's neuromuscular performance changes and adapts over an extended period of time, and therefore may be shortsighted, or even misleading. When assessing athletes' neuromuscular performance via CMJ, practitioners are therefore encouraged to closely monitor how different force-time characteristics change longitudinally.

## Author Contributions

**Conceptualization:** Nicolas M. Philipp, Dimitrije Cabarkapa.

**Data curation:** Nicolas M. Philipp, Dimitrije Cabarkapa, Ramsey M. Nijem.

**Formal analysis:** Nicolas M. Philipp.

**Methodology:** Nicolas M. Philipp, Dimitrije Cabarkapa, Ramsey M. Nijem, Andrew C. Fry.

**Visualization:** Nicolas M. Philipp.

**Writing – original draft:** Nicolas M. Philipp, Dimitrije Cabarkapa.

**Writing – review & editing:** Nicolas M. Philipp, Dimitrije Cabarkapa, Ramsey M. Nijem, Andrew C. Fry.

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
