## [Decision Letter · Decision Letter 0]

27 Dec 2022

PONE-D-22-31306Changes in Countermovement Jump Force-Time Characteristic in Elite Male Basketball Players: A Season-Long AnalysesPLOS ONE

Dear Dr. Nicolas M Phillipp, 

Thank you for submitting your manuscript to PLOS ONE. After careful consideration, we feel that it has merit but does not fully meet PLOS ONE’s publication criteria as it currently stands. Therefore, we invite you to submit a revised version of the manuscript that addresses the points raised during the review process.

ACADEMIC EDITOR:

Dear authors, please find attached the comments of the two reviewers who recommended minor revisions. Specifically, pay attention to the comments regarding reorganization of the information within the manuscript for increasing readability (proposed by the second reviewer). 

We look forward to receiving your revised manuscript.

Kind regards,

Danica Janicijevic, Ph.D

Academic Editor

PLOS ONE

Journal Requirements:

"NO"

"NO"

Reviewers' comments:

Reviewer's Responses to Questions

**Comments to the Author**

1. Is the manuscript technically sound, and do the data support the conclusions?

Reviewer #1: Partly

Reviewer #2: Partly

2. Has the statistical analysis been performed appropriately and rigorously? 

Reviewer #1: Yes

Reviewer #2: Yes

3. Have the authors made all data underlying the findings in their manuscript fully available?

Reviewer #1: Yes

Reviewer #2: Yes

4. Is the manuscript presented in an intelligible fashion and written in standard English?

Reviewer #1: Yes

Reviewer #2: No

5. Review Comments to the Author

Reviewer #1: line 46 -47 need a reference

line 71 - 73 I suggest updating the definition. (10.1519/SSC.0b013e3181e928f9,

10.1016/j.jbiomech.2020.110136)

table 1: the table is not clear, the 57 evaluations correspond to the evaluations of the 12 athletes participating in the study during 7 days of testing? if so, shouldn't it be 84?

and likewise for the other periods. I suggest an explanatory or explanatory note at the bottom of the table.

line 122: mention the code

line 142: this is repeating what is already in table 1

line 156: I suggest a quote that supports this decision check this if it works for you

10.21500/20112084.844

the quality of the images is not the best, try to improve the resolution

Reviewer #2: General comment:

The background of the authors as applied researchers are noticeable in their current work. Some revisions are needed to make it more scientific. In addition, there is lacking information in regard to phase periodization models (tactical periodization during in-season vs. traditional) and training parameters (duration; frequency; mode) which may help justify the findings of the study.

Specific comments:

-avoid using athlete's and similar wordings (CMJ's, team's, etc...) in the manuscript

-make statements more simple

LN 54-66: Re-organize in a way that it highlights the changes in jump mechanics across training phases. Use the lack of studies in real-world settings in the last part to justify the need for this kind of study.

LN 67-69: Paraphrase into more scientific stream.

LN 70: change hopping to jumping.

LN 78-80: just use force platform from this point onwards. Integrate LN 80-81.

LN 125: is the dynamic warm-up consistent across training phases? Expound.

LN 208-260: Too much information presented here. Breakdown into two paragraphs (1: key findings; 2: rationale for results).

REFERENCE: double check and update for adherence with the journal standard. For example:

LN: 367-370: abstract?

LN 372-374: Should be 2022

LN 379: Should be Journal of Exercise in Sports and Medicine

LN 381: Should be Journal of Strength and Conditioning Research

6. PLOS authors have the option to publish the peer review history of their article (what does this mean?). If published, this will include your full peer review and any attached files.

Reviewer #1: **Yes: **Brayan Patiño-Palma

Reviewer #2: **Yes: **Jeffrey Cayaban Pagaduan

---

## [Author Response · Author response to Decision Letter 0]

9 Jan 2023

Specific Comments and Author Response for Reviewer #1:

Line 46-47

“Need a reference”.

 Authors have added a reference to the respective location. Thank you for your constructive feedback. 

Line 71-73

“I suggest updating the definition.”

 Dear reviewer, thank you for your comment. We have eliminated our definition and have cited a definition from the article proposed by yourself. 

Table 1

“The table is not clear, the 57 evaluations correspond to the evaluations of the 12 athletes participating in the study during 7 days of testing? if so, shouldn't it be 84?

and likewise for the other periods. I suggest an explanatory or explanatory note at the bottom of the table”

 We have added a sentence in the paragraph above to clarify what is meant by test days and athlete screenings to avoid any potential confusion. We hope this clarifies things. Given the real-world nature of our data, not all 12 athletes performed the jumps on every single test day due to various reasons (e.g., sickness, class conflicts etc.). We account for these “holes” in the data by using linear mixed effect models as our statistical approach. 

Line 122

“Mention the code.”

 Thank you, we have added the respective IRB approval number.

Line 142

“This is repeating what is already in table 1”.

Dear reviewer, thank you for your constructive feedback. We have eliminated the respective sentence since the information is already provided in table 1.

Line 156

“I suggest a quote that supports this decision”. 

 Dear reviewer, thank you for your comment, we have added an addition to the sentence, as well as the citation.

Figures

the quality of the images is not the best, try to improve the resolution

 Dear reviewer, we believe the quality of the images (JPEG) that were initially submitted are of good quality, however the quality might have been influence in the conversion to a PDF. In this case we would like to refer to the editor for a final decision. If indeed the quality of our images needs improvement, we’d be happy to look into this and make respective changes. 

Thank you, Reviewer #1, for your comments and suggestions. By addressing your prompts, the manuscript should better appeal to readership. 

Specific Comments and Author Response for Reviewer #2:

LN 54-66

“Re-organize in a way that it highlights the changes in jump mechanics across training phases. Use the lack of studies in real-world settings in the last part to justify the need for this kind of study”

 Dear reviewer, thank you for your constructive feedback. We have reordered the paragraph in a way where now, the respective studies are mentioned in the beginning of the paragraph, and the rationale behind why studies in real-world settings are needed, towards the end of the paragraph. We hope this paragraph now reads clearer. 

LN 67-69

“Paraphrase into more scientific stream”. 

 Highlighted in blue, we have made respective changes to the proposed location.

LN 70

“Change hopping to jumping”

 Thank you, we have made the proposed change.

LN 78-80

“Just use force platform from this point onwards. Integrate LN 80-81.”

 Dear reviewer, thank you for this comment, we have made the proposed change in our manuscript. 

LN 125

“Is the dynamic warm-up consistent across training phases?

 Dear reviewer, thank you for your constructive feedback. We have added a note clarifying that the warmup procedures stayed consistent over the duration of the study.

LN 208-260

“Too much information presented here. Breakdown into two paragraphs (1: Key findings; 2: Rationale for results).”

 Dear reviewer, thank you for your comment. We have reduced our discussion, to make it more concise. However, in line with PlosOne’s guidelines, especially in regard to interpreting the results, and how they relate to the hypothesis, and previously conducted research, we fear that by eliminating further sections from the discussion, our manuscript loses strength. If in its current standing, the discussion section still does not fulfill what is expected for publication, we kindly ask you, as well as the editorial team to make suggestions slightly more detailed, in order for us authors to edit specific parts of the discussion. We'd be happy to incorporate further changes if suggested. We greatly appreciate your feedback and believe our manuscript will be improved based on your comments. 

REFERENCES

 Thank you, we have made suggested changes to our references, and highlighted these in blue. 

Thank you, Reviewer #2, for your comments and suggestions. By addressing your prompts, the manuscript should better appeal to readership.

---

## [Editor Report · Decision Letter 1]

19 May 2023

Changes in Countermovement Jump Force-Time Characteristic in Elite Male Basketball Players: A Season-Long Analyses

PONE-D-22-31306R1

Dear Dr. Philipp,

We’re pleased to inform you that your manuscript has been judged scientifically suitable for publication and will be formally accepted for publication once it meets all outstanding technical requirements.

Kind regards,

Goran Kuvačić, PhD

Academic Editor

PLOS ONE

Additional Editor Comments (optional):

Reviewers' comments:

<quillbot-extension-portal></quillbot-extension-portal>

---

## [Editor Report · Acceptance letter]

24 May 2023

PONE-D-22-31306R1 

Changes in Countermovement Jump Force-Time Characteristic in Elite Male Basketball Players: A Season-Long Analyses 

Dear Dr. Philipp:

I'm pleased to inform you that your manuscript has been deemed suitable for publication in PLOS ONE. Congratulations! Your manuscript is now with our production department. 

Kind regards, 

on behalf of

Dr. Goran Kuvačić 

Academic Editor

PLOS ONE